# Non-Uniform Curvature Model and Numerical Simulation for Anti-Symmetric Cylindrical Bistable Polymer Composite Shells

**DOI:** 10.3390/polym12051001

**Published:** 2020-04-26

**Authors:** Zheng Zhang, Xiaochen Yu, Helong Wu, Min Sun, Xianghao Li, Huaping Wu, Shaofei Jiang

**Affiliations:** Key Laboratory of E&M, Zhejiang University of Technology, Ministry of Education & Zhejiang Province, Hangzhou 310014, China; zzhangme@zjut.edu.cn (Z.Z.); yxc_zgd@163.com (X.Y.); helongwu@zjut.edu.cn (H.W.); ryanli2810@gmail.com (X.L.); hpwu@zjut.edu.cn (H.W.); jsf75@zjut.edu.cn (S.J.)

**Keywords:** bistability, anti-symmetric cylindrical shell, polymer composite, non-uniform curvature, finite element simulation, python

## Abstract

The bistability of anti-symmetric thin shallow cylindrical polymer composite shells, made of carbon fiber/epoxy resin, has already been investigated based on the uniform curvature and inextensible deformation assumptions by researchers in detail. In this paper, a non-uniform curvature model that considers the extensible deformations is proposed. Furthermore, a parametric modeling and automatic postprocessing plug-in component for the bistability analysis of polymer composite cylindrical shells is established by means of ABAQUS-software, by which the equilibrium configurations and the load-displacement curves during the snap process can be easily obtained. The presented analytical model is validated by the numerical simulation and literature models, while the factors affecting the bistability of anti-symmetric cylindrical shells are revisited. In addition, the planform effects of anti-symmetric cylindrical shells with rectangular, elliptical and trapezoidal planform are discussed. The results show that the presented analytical model improves the accuracy of the prediction of the principal curvature of second equilibrium configuration and agree well with the numerical results.

## 1. Introduction

Owing to the ability to provide stiffness and strength during large shape change, bistable thin shallow polymer composite structures have shown potential as morphing structures used in aerospace [1,2,3], origami [4,5] and bionics [6,7,8]. Such morphing structure can also find applications in vibration energy harvest systems [9,10] and gearless motors [11], as well as anti-icing systems [12]. The bistability may stem from the pre-stress or plastic deformations [13,14,15,16], the inducing stresses through thermal effects for non-symmetric layup [17,18,19], or the combined influence of the material properties and geometry [20,21,22]. Since the equilibrium configurations can be quite different under various manufacturing methods, it demands that the analytical model and finite-element analysis are able to predict the bistability. 

In this paper, we focus on the anti-symmetric cylindrical shells that are made of carbon-fiber/epoxy resin composites. Based on the assumption of uniform curvature and the classical laminate theory (CLT), some simplified models [23,24,25,26] have been proposed assuming the inextensible deformation, which implies that the second equilibrium configuration remained as a standard cylinder. A two-parameter inextensional uniform curvature model (IUCM) used to analyze the bistability of cylindrical shells was summarized and proposed by Guest and Pellegrino [20]. On this basis, the factors affecting the bistability have been systematically investigated [27,28,29,30,31], showing a reasonable agreement among experimental, numerical and analytical results. However, the inextensible deformation assumption may lead to inconformity with the practical deformations. Thus, the accuracy of the analytical prediction deserves further consideration.

Considering the importance of initial Gaussian curvature [32], which further governs the relative variation between the stretching and bending strain energy densities, Seffen [21] put forward an extensible uniform curvature model (EUCM). It considered the practical deformations of shells instead of that in inextensible deformation assumption. The combined influence of the material properties and geometry was explored in a systematic way to reveal possible bistable solutions. On this basis, Vidoli and Maurini [22] extended the EUCM, and found the existence of three equilibrium configurations for a specific wide range of initial curvatures. Furthermore, the inelastic deformations pertinent to thermal, hygroscopic or plastic effects were summarized and introduced [33]. The multistability of shells was studied to provide the basic criteria for the design of multistable shells [34]. In addition, Vidoli [35] developed non-uniform curvature models that assumed curvature to vary linearly and quadratically, labeled as linearly varying curvature model (LCVM) and quadratically varying curvature model (QCVM), respectively. The bistability of unsymmetric plates was analyzed and the dependence of the average curvatures on the aspect ratio was correctly captured. The bistable equilibrium configurations of thin shallow shell with clamped boundary condition [36,37] were predicted as well. The uniform curvature assumption was also weakened [38,39] by using high-order approximations for displacement field on the basis of the Rayleigh–Ritz method and the discretization of Föppl–Von Kármán (FVK) equations [40]. Thus, complex nonlinear bistable behavior can be captured in detail by using high-order approximations and the non-uniform curvature assumption despite of a certain computational cost.

To the best of our knowledge, the bistability of anti-symmetric cylindrical polymer composite shells has not been reported based on the non-uniform curvature assumption. This paper adopts extensible deformation assumption and presents a non-uniform curvature model (NUCM), assuming curvatures to vary quadratically as well, for more accurate predictions. The layout of this paper is as follows: in Section 2, the NUCM is proposed. The presented analytical model is verified by fully non-linear finite element analysis (FEA) using an ABAQUS plug-in component based on Python script. The specific plug-in component allowing for parameterized geometric modeling and automatically postprocessing in a systematic and efficient way is introduced in Section 3. Section 4 validates the accuracy of the presented model. In addition, the factors affecting the bistability of anti-symmetric cylindrical polymer composite shells are discussed, especially the planform effects of them. Section 5 draws the conclusions.

## 2. Analytical Model

The analytical model follows the similar reduction procedure of Vidoli [35]; however, several extensions have been incorporated. The main features of the presented model are briefly recalled. One can see Appendix A for more details.

The stable equilibrium configurations of an FVK shell can be found as local minima of the total energy. The shells’ boundaries are free without applied forces, and the total potential energy is expressed as [35]
(1)$$\bar{U}=\frac{1}{2}\int_{\Omega }[\bar{D}(K-H)\cdot (K-H)+\frac{12R_{0}^{2}}{t_{z}^{2}}{\bar{A}}^{-1}\Sigma \cdot \Sigma ]d\Omega$$

Equation (1) is a dimensionless form and the dimensionless quantities are as follows
(2)$$\begin{array}{l} X=x/L,Y=y/L,W=w/L,K=R_{0}k,\;H=R_{0}h\\ \bar{A}=A/A_{11}^{},\bar{B}=B/(A_{11}^{}R_{0}),\bar{D}=D^{*}/D_{11}^{*},\Sigma =N/A_{11}^{}\\ \bar{U}=UR_{0}^{2}/(L^{2}D_{11}^{*}) \end{array}$$
where *L*^2^ is the area of the planform of the shell, *R*_0_ is a characteristics radius employed to scale the curvatures, *N* is the membranal stress, and *w* is the tranverse displacement. ***H*** represents the dimensionless initial curvature and describes the initial stress-free configuration. ***K*** represents the dimensionless curvature. ***A***, ***B***, ***D*** are 3 × 3 matrices representing the extensional stiffness, the extension-to-bending coupling, and the bending stiffness, respectively. Specifically, ***D***^*^ = ***D*** − ***B***^T^***A***^−1^***B***. Under the consideration of the homogeneous deformation of shell through the thickness, the equivalent thickness is given as $t_{z}=\sqrt{12D_{11}^{*}/A_{11}}$.

Considering the orthotropic material, the dimensionless stiffnesses can be written as
(3)$$\begin{array}{l} \bar{A}=\left[\begin{array}{lll} 1 & v_{a} & 0\\ v_{a} & \beta_{a} & 0\\ 0 & 0 & \gamma_{a} \end{array}\right],\;\bar{D}=\left[\begin{array}{lll} 1 & v_{d} & 0\\ v_{d} & \beta_{d} & 0\\ 0 & 0 & \gamma_{d} \end{array}\right]\\ \gamma_{a}=\rho_{a}(1-v_{a}^{2}/\beta_{a}),\;\gamma_{d}=\rho_{d}(1-v_{d}^{2}/\beta_{d}) \end{array}$$
where *v_a_* measures in-plane Poisson effect, *β_a_* measures the ratio between the extensional stiffnesses in the coordinate directions, *ρ_a_* is the shear modulus, *v_d_* measures the Poisson effect in bending, *β_d_* measures the ratio between the bending stiffnesses in the coordinate directions, and *ρ_d_* is the torsional modulus. 

Substitute the constitutive relation of orthotropic material into the Gaussian compatibility (Equation (A1), see Appendix A):(4)$$\frac{\beta_{a}-v_{a}}{\beta_{a}-v_{a}^{2}}\frac{\partial^{2}\Sigma_{X}}{\partial Y^{2}}+\frac{1-v_{a}}{\beta_{a}-v_{a}^{2}}\frac{\partial^{2}\Sigma_{Y}}{\partial X^{2}}-\frac{2}{\gamma_{a}}\frac{\partial^{2}\Sigma_{XY}}{\partial X\partial Y}=K_{x}K_{y}-K_{xy}^{2}-(H_{x}H_{y}-H_{xy}^{2})=\Delta G$$
where Δ*G* is the difference in dimensionless Gaussian curvature between the actual and natural configurations. The Gaussian curvatures are determined by the dimensionless transverse displacement *W*. The uniform curvature assumption represents that the Gaussian curvature is constant. The quadratically varying curvature assumption is adopted in NUCM, representing the Gaussian curvature, which varies quadratically. Cylindrical shells with three different planform shapes are discussed, labeled as rectangular planform shells (RPS), elliptical planform shells (EPS) and trapezoidal planform shells (TPS), respectively (see Figure 1). The coordinates are chosen to be centered in the centroid of the planform. For an RPS, the lengths along the *x*- and *y*- directions are 2*a* and 2*b*, while for an EPS, the major and minor axes lengths are 2*a* and 2*b*. Moreover, the TPS can be obtained by tailoring the shell with rectangular planform, in which an additional parameter *c* is used. The transverse displacements for RPS and EPS are expressed, respectively, as
(5)$$W=\frac{1}{2}s_{1}X^{2}+\frac{1}{2}s_{2}Y^{2}+s_{3}XY+s_{4}X^{2}(X^{2}-\frac{1}{2r})+s_{5}Y^{2}(Y^{2}-\frac{r}{2})$$
(6)$$W=\frac{1}{2}s_{1}X^{2}+\frac{1}{2}s_{2}Y^{2}+s_{3}XY+s_{4}X^{2}(\frac{X^{2}}{3}-\frac{1}{2\pi r})+s_{5}Y^{2}(\frac{Y^{2}}{3}-\frac{r}{2\pi })$$
which satisfy the average curvatures over the shell to be constant. In fact, *s*_1_, *s*_2_, *s*_3_ control the average curvatures over the shell, whilst *s*_4_ and *s*_5_ control the quadratic variation of the curvature along the direction *x* and *y*, respectively. *r* represents the aspect ratio, namely *r* = *b*/*a*. Due to the difficulty in choosing a transverse displacement to satisfy the average curvatures over the shell to be constant, the transverse displacement for TPS is chosen to be the same as that for RPS. 

By substituting the corresponding transverse displacements into Equation (4), Δ*G* and the stress fields ***Σ*** for each *W* can be written in a same form, given by
(7)$$\begin{array}{l} \Delta G=S_{1}+S_{2}X^{2}+S_{3}Y^{2}+S_{4}X^{2}Y^{2}\\ \Sigma =\frac{l^{2}}{R_{0}^{2}}[S_{1}T_{1}+S_{2}T_{2}+S_{3}T_{3}+S_{4}T_{4}] \end{array}$$
where the detail of *S_i_*, ***T****_i_* (*I* = 1,2,3,4) can be seen in Appendix A.

Therefore, the stretching energy can be obtained by using Equation (7) and the total potential dimensionless energy is then given as
(8)$$\bar{U}=\frac{1}{2}\int_{\Omega }\bar{D}(K-H)\cdot (K-H)d\Omega +\frac{1}{2}\sum_{i=1}^{4}\sum_{j=1}^{4}{\bar{S}}_{ij}\psi_{ij}$$
where ${\bar{S}}_{ij}=S_{i}S_{j},\psi_{ij}=\int_{\Omega }{\bar{A}}^{-1}T_{i}:T_{j}d\Omega \;(i,j=1,2,3,4)$. The scalar factor *ψ_ij_* measures the ratio between bending and stretching energy and is obtained based on FEniCS framework [41]. 

Based on the principle of virtual work and the theorem of minimization of the total potential energy, the first variation of the total energy should be zero [42]
(9)$$f_{i}=\partial \bar{U}/\partial s_{i}=0\;\;\;(i=1,2,\ldots ,5)$$

Thus, the unknown coefficient *s_i_* can be obtained by solving Equation (9), by which the equilibrium configurations are determined. The average curvature over the shell is used to describe the second equilibrium configuration of anti-symmetric cylindrical shell. The stability of the equilibrium configurations can be assessed by examining positive definiteness of the Jacobian matrix [43]
(10)$$J=\frac{\partial f_{i}}{\partial s_{j}},\;\;i,j=1,2,\ldots ,5$$

In the context of this work, the expressions for *f_i_* and *J* are computed symbolically via MATHMATICA software and are solved with the aid of the MATLAB programming.

## 3. Finite Element Analysis

The bistability analysis of anti-symmetric cylindrical shells is validated by finite element simulation performed on ABAQUS software (version 6.14-1, Dassault Systemes Simulia Corp., Providence, RI, USA). Basically, the cylindrical shell can be characterized by the longitudinal length *l*, the angle of embrace *γ*, the initial curvature *h*_0_ and the planform shapes (see Figure 2a). The cylindrical shell with other planform shapes can be obtained by tailoring shell with rectangular planform. The material parameters used in the subsequent numerical simulations are listed in Table 1. The layup parameters consist of the layup angle and the thickness of a single layer *t*. The S4R shell element is chosen for a better convergence. 

Two loading methods, the four-point loading method [44] and the two-point loading method [19], are adopted in numerical simulation to induce the snap-through and snap-back of anti-symmetric cylindrical shells. The four-point loading method is realized by imposing suitable displacements loads in four points at the shell edges intersecting the *xz*- and *yz*- planes and keeping the center of shell totally restrained, as indicated in Figure 2a, which has been used to predict the tristability of orthotropic doubly curved shell [44]. The two-point loading method is realized by moving the rigid indenter downward while keeping the two smooth supporting platforms fixed and the center of shell restrained, as shown in Figure 2b. It is an experimental method for the transformation of anti-symmetric cylindrical shell, which was also adopted in previous literature [27,28,29,30]. The load-displacement curves can be obtained by the indenter in finite element model in contrast to experimental results. Two steps are included in simulation: (1) the displacement loadings are applied with the option *Nlgeom* on and *stabilization with dissipated energy fraction* on with default values; (2) the displacement loadings are removed with the option *Nlgeom* on, and *stabilize* off to avoid inaccuracies. Thus, the cylindrical shell remains in a stress-free state with only the center of shell restrained, and will then transform to the second equilibrium configuration (see Figure 2c,d).

In the postprocessing, the curvatures are used to depict the equilibrium configurations, which are the average of the local curvatures for the whole elements. This conforms to the theoretical assumption. The load-displacement curves are obtained from a reference point of the indenter for the two-point loading method or two different loading points in the shell for the four-point loading method, thus the snap loads of the anti-symmetric cylindrical shells based on different loading method are obtained. Eventually, all of the above messages can be exported in a *csv* file for later usage.

The simulation procedure of the anti-symmetric cylindrical shells is summarized by using a Python script. Then, a parametric modeling plug-in is established based on the Python script to make the anti-symmetric cylindrical shell parametrically generated and the postprocessing automatically completed. The process of the plug-in is shown in Figure 2. The downward distance *ldown* and the gap between supporting platforms *lgap* in the two-point loading method and the downward and upward distance *down* and *up* in the four-point loading method can be adjusted for a better convergence. The default values are empirically set as: *ldow*n = 1/*h*_0_ + 15 mm, *lgap* = 50 mm, based on the experimental observations, and *down* = 1/*h*_0_, *up* = 0.4*l*, based on the trial simulation.

## 4. Result and Discussion

In this section, results are presented with examples of an anti-symmetric cylindrical shell. The results from NUCM are compared with the results from FEA, IUCM, EUCM and QVCM. Unless otherwise stated, the geometric parameters of the anti-symmetric cylindrical shell are *l* = 120 mm, *h*_0_ = 0.04 mm^−1^, *γ* = 180°, *t* = 0.12 mm with a rectangular planform and the layup is [45°/−45°/45°/−45°].

### 4.1. Validation of Accuracy

The comparison results are demonstrated in Table 2, where *k_x_*_2_, *k_y_*_2_ represent the average curvatures in second equilibrium configuration of anti-symmetric cylindrical shells. 

Compared with the QVCM with two degrees of freedom, the NUCM has five degrees of freedom, leading to the increase in computational cost and the complexity of the expression. But a large deviation is observed in Table 2 for QVCM against the numerical results. The same results are observed in many other examples that are therefore omitted here for brevity. It can be concluded that the QVCM, satisfying the bending boundary conditions on average, is not suitable to predict the bistability of an anti-symmetric cylindrical shell and will not be discussed later. In contrast, the results from EUCM and NUCM have a good agreement with the numerical results. In addition, compared to the results from EUCM, significant improvement in the accuracy of the average curvature is observed. The relative errors of the principal curvature *k_x_*_2_ are 1.04%, 10.11% and 11.15%, respectively, for NUCM, EUCM and IUCM. A remarkable improvement in accuracy is also observed for the curvature *k_y_*_2_. However, it is less important than the principal curvature *k_x_*_2_ and is neglected in the following discussion.

### 4.2. Factors Affecting the Bistability

The factors affecting the bistable behavior of anti-symmetric cylindrical shells are investigated by the NUCM and FE simulation. The results from EUCM and IUCM are also given for a direct comparison. The bistable performances of anti-symmetric cylindrical shells are mainly affected by: (i) layup parameters containing angle of layup α, number of plies *p*, (ii) geometrical parameters including the angle of embrace *γ*, the longitudinal length *l*, the initial nature curvature *h*_0_ and planform shapes. Some factors have been studied in the previous literature [28], but a few extra considerations are reported with the NUCM. In essence, these factors directly influence the bistability of an anti-symmetric cylindrical shell in numerical model, but yet some of which do not affect straightforwardly in the NUCM. The direct factors for the bistability of anti-symmetric cylindrical shell in NUCM are: (i) the geometrical parameters used to describe the planform shape with the initial nature curvature, (ii) the dimensionless stiffness parameters, (3) the scalar factor *ψ**_ij_* and characteristic radius *R*_0_ related to the dimensional transformation, both of which are affected by the former two factors. The relationships of the direct factors between the numerical simulation and NUCM are shown in the Figure 3. The geometric parameters relationship between them are: *a* = *l*/2, *b* = sin(*γ*/2)/*h*_0_, in which *γ* cannot exceed 180 degrees due to the duplication of *b*. Three examples are selected, referred to as No.1, No.2 and No.3, respectively. No.1 represents the classical anti-symmetric cylindrical shell as stated before. No.2 and No.3 are the same as No.1 except for a change in layup parameters (replaced with *p* = 5) or geometric parameters (replaced with *a* = 21.56 mm), respectively.

#### 4.2.1. Influence of Scalar Factor and Characteristic Radius

The scalar factor *ψ**_ij_* is influenced by the layup parameters, material parameters and the geometrical parameters except for the initial curvature. When the scalar factor is determined, the bistability of anti-symmetric cylindrical shells can be investigated with the variation of dimensionless curvatures of the natural stress-free configuration (the initial twist curvature is assumed to be zero), (see bistability curves in Figure 4). When the characteristic radius *R*_0_ is determined, the principal curvature *k_x_*_2_ can be determined through dimensional transformation, corresponding to a point, i.e., the black dot shown in the Figure 4. All the bistability curves show similar trends with the different critical value, which relates to the vanishing of bistability. The scalar factor changed by geometrical parameters shows a slightly deviation for the trend and the initial dimensionless curvature *H*, but an obvious difference in the dimensionless curvature *K_X_*_2_. It is contrary to the scalar factor changed by layup parameters. In addition, the bistability curves of NUCM tend to be lower than that of EUCM, and the critical value is greater, which conforms to the trend shown in the Figure 9 in Ref. [35]. The bistability of anti-symmetric cylindrical shells exists when the point is on the curve, and disappears when the point deviates from the curve. Note that the corresponding point and the bistability curve vary for different cases. In this sense, the bistability range predicted by NUCM is narrower than that by EUCM. It can be concluded that the scalar factor *ψ**_ij_* and characteristic radius *R*_0_ are the most important direct factors in the NUCM affecting the bistability of anti-symmetric cylindrical shells. 

#### 4.2.2. Influence of Layup Parameters

The influence of the angle of layup α is investigated by varying α in the layup of [α/−α/α/−α], (see Figure 5). The IUCM, EUCM, NUCM and FEA predict that the bistability exists at the layup angle ranging from 30° to 60°, 30° to 60°, 34° to 58°and 33° to 51°, respectively. All of them show a gradually increasing trend of principle curvature *k_x_*_2_ and demonstrate that the angle of layup has significant influence on the bistability of anti-symmetric cylindrical shells. The results from NUCM agree well with FE results within the overlap range of layup angle. The discrepancy between results of NUCM and FEA reaches the minimum value 1% at α = 45°, and increases as α is drawn apart from 45°, not exceeding 10%.

The results of cases with *p* from 4 to 8 are shown in Figure 6. The influence of the number of plies *p* on principal curvature *k_x_*_2_ is based on the layup with α = 45°. It should be noted that the ply angle of the middle ply is zero when *p* is odd. For example, when *p* = 4 and 5, the layups are [α/−α/α/−α] and [α/−α/0°/α/−α], respectively. In these two cases, the total thickness is the same, but the thickness of each single layer is different. It is seen in Figure 6 that the analytical and FE results show a similar trend and the number of plies *p* has a little influence on the principal curvature *k_x_*_2_. The discrepancy between the results of NUCM and FEA is approximately within 1%, as seen in Table 3, where the error represents the relative error between results from FEA and NUCM.

Additionally, the influence of material parameters has not been discussed because they affect the bistability of anti-symmetric cylindrical shell through the dimensionless stiffness parameters, the same as layup parameters do (see Figure 3).

#### 4.2.3. Influence of Geometrical Parameters

Figure 7 shows the influence of the angle of embrace *γ* and comparison the analytical and FE results. For IUCM, the principal curvature *k_x_*_2_ are constant irrespective of the variation of *γ* and the disappearance of bistability is unable to be predicted. As the extensible deformation assumption applied, the disappearance of bistability can be predicted, with the critical value of *γ* = 63°, 88° and 94° for EUCM, NUCM and FEA, respectively, and then the principal curvature *k_x_*_2_ gradually increases with the increase in *γ* (see Figure 7). The results from NUCM and FEA agree well with the relative error below 5%, as seen in Table 4 where the error represents error between results from FEA and NUCM.

For IUCM, the principal curvature *k_x_*_2_ are constant, irrespective of the variation of *l* and the disappearance of bistability is unable to be predicted as well. As the extensible deformation assumption is applied, the influence of the longitudinal length *l* on the principal curvature can be predicted, as shown in Figure 8, with the critical values of *l* = 30, 40 and 32 mm for EUCM, NUCM and FEA, respectively. In fact, the anti-symmetric cylindrical shell with sufficiently large length tends to roll up in the second equilibrium configuration, and thus *k_x_*_2_ tends to be a constant. As the increase in *l* occurs, the principal curvatures *k_x_*_2_ of EUCM and FEA show a similar trend and tend to be constant when the length is over 60 and 100 mm, respectively. In contrast, the *k_x_*_2_ of NUCM gradually increases and converges at a length greater than 160 mm. Nonetheless, difference between the results of NUCM and numerical results (1.0 m^−1^ on average) is closer to that between the results of EUCM and numerical results (2.6 m^−1^ on average), as shown in Figure 8.

The influence of the initial natural curvature *h*_0_ is shown in Figure 9 in which the initial natural curvature changes while keeping the aspect ratio *r* unchanged. According to IUCM, the principal curvature *k_x_*_2_ can be simply calculated as *k_x2_ = v_d_h_0_*. In addition, as the initial curvature increases, the principal curvature *k_x_*_2_ grows, with the critical values of *h*_0_ = 0.140, 0.085 and 0.137 mm^−1^ for EUCM, NUCM and FEA, respectively. A sensible increment of the principal curvature is observed in Figure 9 for numerical results and results obtained by EUCM and NUCM, which show the disappearance of bistability. It can be seen that the numerical results agree well with the results obtained by NUCM for small initial natural curvature, as listed in Table 5 where the error-1 and error-2 represent the relative error between the results of FEA and EUCM and that of FEA and NUCM, respectively. Whilst for large initial natural curvature the results obtained by EUCM are more reliable in terms of the relative error and critical value. The possible reason for the increasing error is that a large value of *h*_0_ may violate the thin shallow shell assumption [34] when the total thickness is unchanged.

The influence of the anti-symmetric cylindrical shell with different planform shapes is investigated by EUCM, NUCM and FEA in Table 6, where the error-1 and error-2 represent the relative error between results from FEA and EUCM and that from FEA and NUCM, respectively. Three different planform shapes are considered, as shown in Figure 1. The geometric parameters are: *a* = 60 mm, *b* = 25 mm, *c* = 5 mm. Theoretically, for NUCM, the planform shapes influence the bistability of anti-symmetric cylindrical shells through the integral region in Equation (8) and the scalar factor *ψ_ij_* (*i*,*j* = 1,2,3,4), but only through a single scalar factor *ψ*_11_ for EUCM. With the change in planform shape, the scalar factor *ψ*_11_ changes slightly, resulting in the slightly deviation of the results of EUCM. The decline and increase in principal curvature *k_x_*_2_ are found for results of EPS and TPS from EUCM, respectively, compared to the principal curvature of RPS. However, the contrary results are found based on the FEA results as listed in Table 6, which indicates the lower inaccuracy of EUCM for the prediction of planform effects of anti-symmetric cylindrical shells. Compared to principal curvature of RPS, the increase in principal curvature *k_x_*_2_ of EPS is correctly predicted using NUCM. In addition, a more accurate result is obtained from NUCM in contrast with the result from EUCM, as shown in error-1 and error-2 in Table 6. Although the decline in principal curvature *k_x_*_2_ of TPS is not predicted, the relative error between the results from the analytical model and simulation are reduced by utilizing NUCM, as listed in Table 6. 

The possible reason for the deviation between results of TPS from analytical model may lay on the difference of bending boundary effect for different planform shapes [44,45]. The bending boundary effect develops for any configuration other than the base state and is due to the disequilibrium of the bending moment. The change in local curvature occurs as the bending moment is equilibrated by the out-of-plane shear stress, which results in the change in average curvature. The local curvatures discussed later are absolute values. Due to the bending boundary effect, the local curvatures in the corner and edge of RPS decrease and increase, respectively (see Figure 10a), where A_1_, A_2_, A_3_, A_4_ are labeled as the region near the corner; the local curvatures of EPS decrease near the region B_1_, B_2_, B_3_, B_4_ and increase in the edge between them (see Figure 10b); the local curvatures of TPS decrease obviously near the region C_3_, C_4_ and less obviously near the region C_1_, C_2_ (see Figure 10c). The increase in local curvature of EPS is greater than that of RPS, as shown in Figure 10, which demonstrates the greater average principal curvature *k_x_*_2_ of EPS. The curvature distribution of TPS is similar to that of RPS, and the decrease in local curvature of TPS is apparent, which demonstrates the smaller average principal curvature *k_x_*_2_ of TPS, compared to that of RPS. However, in the NUCM, the transverse displacement for TPS is chosen to be the same as that for RPS for simplicity, which may be the reason why the results of TPS from NUCM are large.

## 5. Conclusions

This paper proposes a non-uniform curvature model with five degrees of freedom to investigate the bistability of the anti-symmetric cylindrical shells. Additionally, a plug-in based on the python script is established through ABAQUS software, with the ability of parameter modeling for the anti-symmetric cylindrical shell with rectangular, elliptical and trapezoidal planforms and automatic postprocessing of the equilibrium configurations and the load-displacement curves during the snap process. The accuracy of the presented model is verified by the finite element simulation and the comparison with EUCM and IUCM. Moreover, the effects of various factors for the bistability of the anti-symmetric cylindrical shells are studied. The results indicate that the angle of layup *α* and the initial natural curvature *h*_0_ are the major factors influencing the principal curvature. The angle of embrace *γ*, the longitudinal length *l* and the number of plies *p* have lesser impact on the principal curvature, not as predicted in IUMC for the former two factors. The planform effects of anti-symmetric cylindrical shells are investigated. The results between analytical model and simulation of RPS and EPS correlate well, whilst a certain error exists in the results between the analytical model and simulation of TPS, as shown in Table 6. The possible reason has been discussed. In this respect, more appropriate approximation polynomials for the transverse displacement field corresponding to each planform shape should be considered. Moreover, higher order or more complex approximation polynomials might be considered for the transverse displacement field so as to overcome the limitation and improve the versatility of the presented model. Further investigations are undergoing to better understand the planform effects on the bistability of anti-symmetric cylindrical shells. 

## Figures and Tables

**Figure 1 polymers-12-01001-f001:**
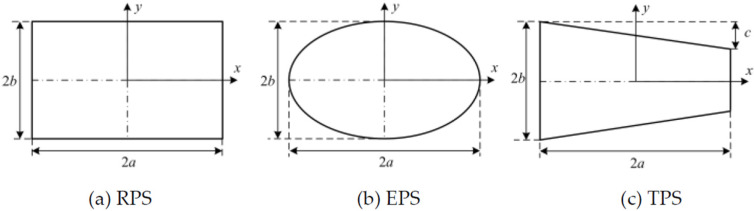
Geometrical parameters of cylindrical shells with different planform shapes.

**Figure 2 polymers-12-01001-f002:**
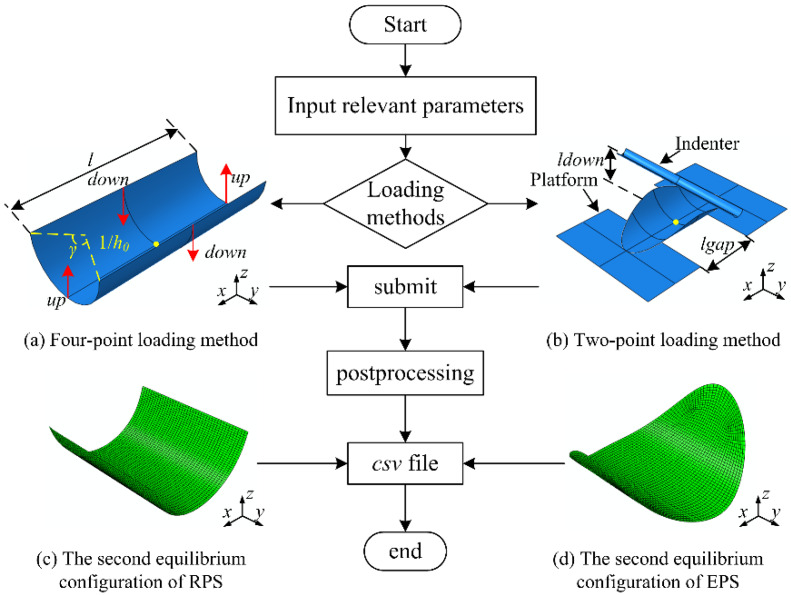
The process of the plug-in for the bistability analysis of anti-symmetric cylindrical shells.

**Figure 3 polymers-12-01001-f003:**
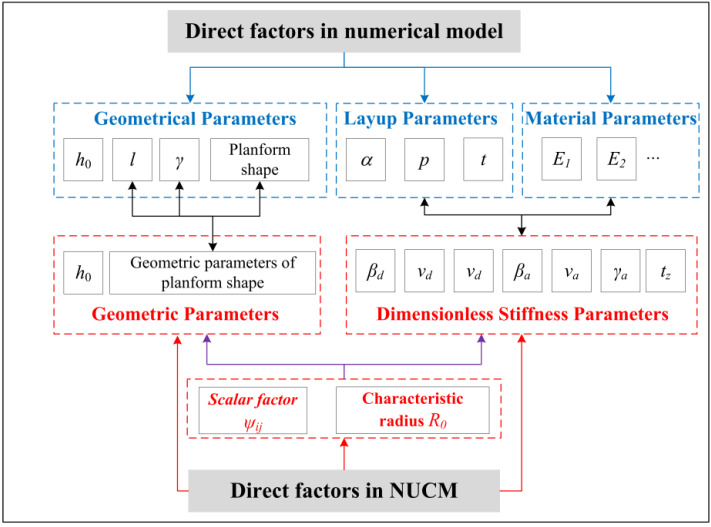
Relationships of the direct factors between the numerical model and non-uniform curvature model (NUCM).

**Figure 4 polymers-12-01001-f004:**
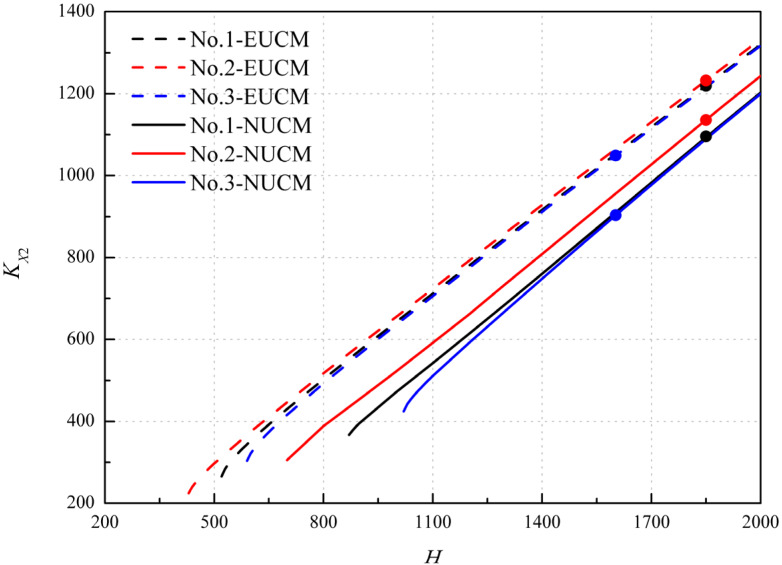
The bistability curves at different scalar factor.

**Figure 5 polymers-12-01001-f005:**
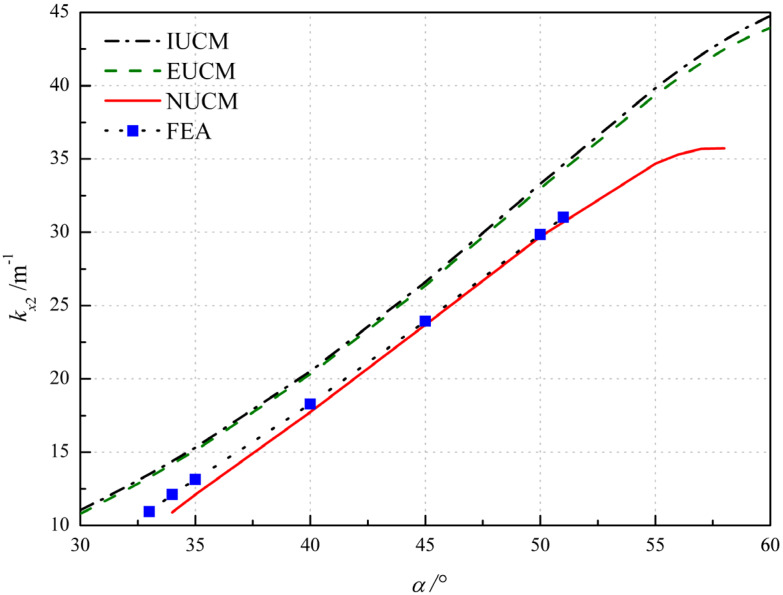
Variation of the principal curvature with respect to the angle of layup α.

**Figure 6 polymers-12-01001-f006:**
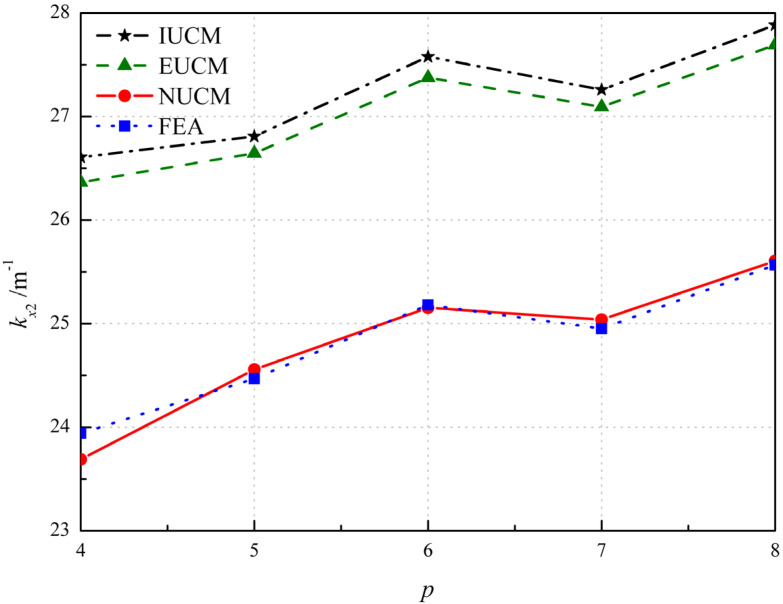
Variation of the principal curvature with respect to the number of plies *p.*

**Figure 7 polymers-12-01001-f007:**
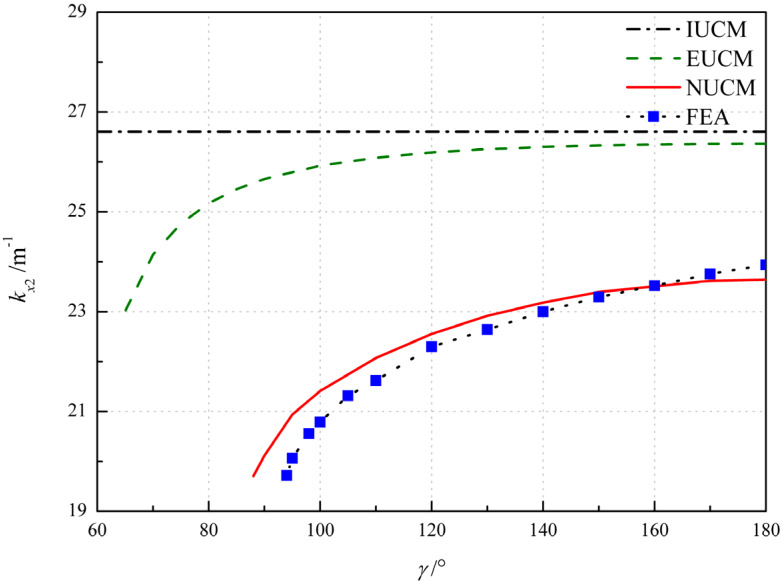
Variation of the principal curvature with respect to the angle of embrace *γ*.

**Figure 8 polymers-12-01001-f008:**
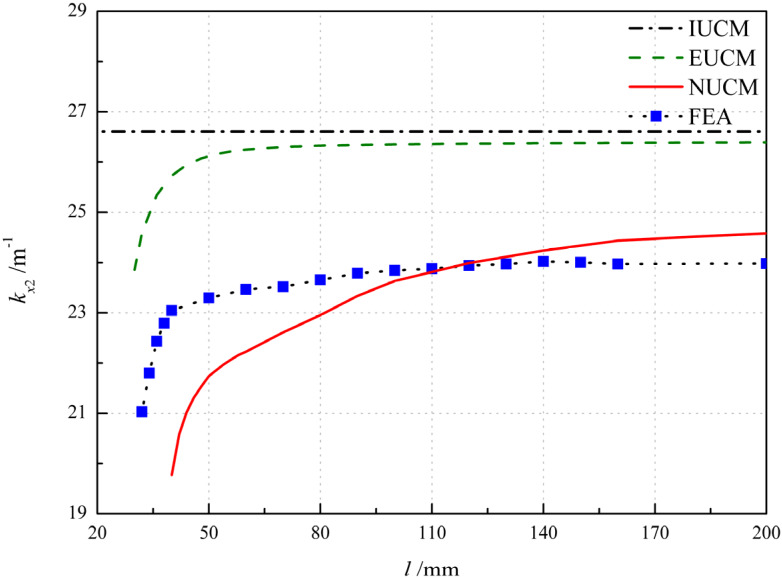
Variation of the principal curvature with respect to the longitudinal length *l.*

**Figure 9 polymers-12-01001-f009:**
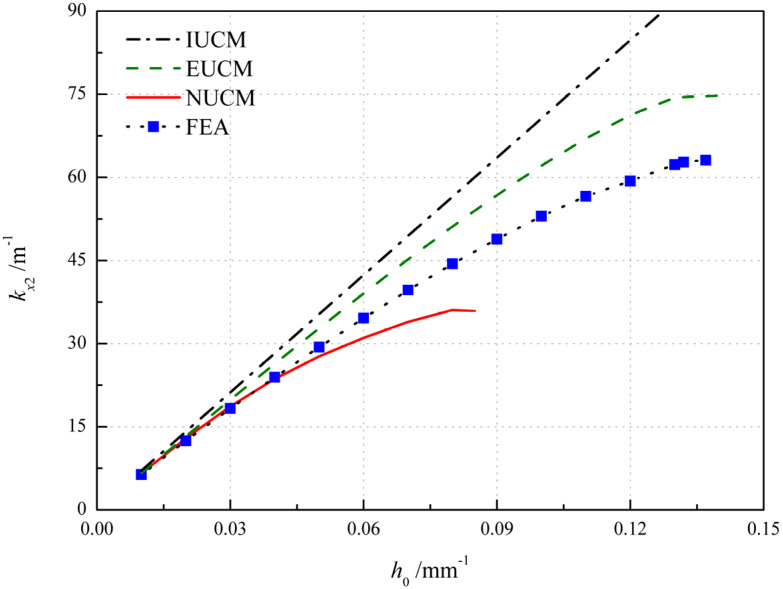
Variation of the principal curvature with respect to the initial natural curvature *h*_0_.

**Figure 10 polymers-12-01001-f010:**
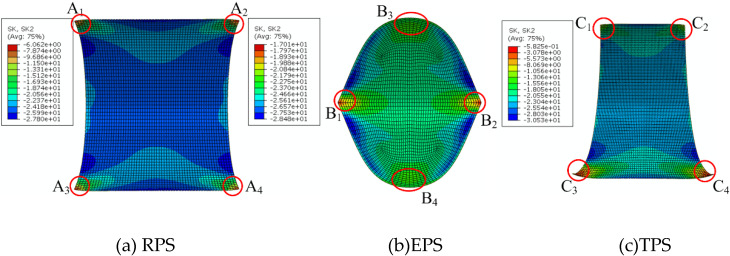
The curvature distribution of *k_x_*_2_ of anti-symmetric cylindrical shells with different planform shapes (vertical view).

**Table 1 polymers-12-01001-t001:** Material parameters of T700/Epoxy resin unidirectional lamina [27].

E_1_/GPa	E_2_/GPa	G_12(13)_/GPa	*v* _12_	α_1_/(10^−6^/°C)	α_2_/(10^−6^/°C)
108	7.07	5.17	0.31	−2.1	64.33

where *E*_1_ and *E*_2_ refer to the longitudinal and transverse elastic modulus, respectively; *ν*_12_ is the in-plane Poisson’s ratio and *G*_12_, *G*_13_ the shear modulus and α_1_, α_2_ are thermal expansion coefficient.

**Table 2 polymers-12-01001-t002:** Comparison of the average curvatures with analytical models and simulation.

Average Curvature	IUCM [20]	EUCM [21]	QVCM [35]	NUCM	FEA
*k_x_*_2_/m^−1^	26.61	26.36	11.98	23.69	23.94
*k_y_*_2_/m^−1^	0	0.16	8.01	2.07	2.81

**Table 3 polymers-12-01001-t003:** Principal curvature of anti-symmetric cylindrical shells with different number of plies from analytical model and simulation.

Number of Plies *p*	*k_x_*_2_/m^−1^(NUCM)	*k_x_*_2_/m^−1^(FEA)	Error
4	23.69	23.94	1.05%
5	24.56	24.47	−0.36%
6	25.15	25.18	0.10%
7	25.04	24.95	−0.35%
8	25.61	25.57	−0.15%

**Table 4 polymers-12-01001-t004:** Principal curvature of anti-symmetric cylindrical shells with different angle of embrace from analytical model and simulation.

Angle of Embrace *γ*/°	*k_x_*_2_/m^−1^(NUCM)	*k_x_*_2_/m^−1^(FEA)	Error
180	23.64	23.94	1.24%
160	23.51	23.52	0.05%
140	23.18	23.00	−0.79%
120	22.55	22.30	−1.14%
100	21.41	20.78	−3.01%
95	20.94	20.06	−4.38%

**Table 5 polymers-12-01001-t005:** Principal curvature of anti-symmetric cylindrical shells with different initial natural curvature from analytical model and simulation.

*h*_0_/mm^−1^	*k_x_*_2_/m^−1^(EUCM)	*k_x_*_2_/m^−1^(NUCM)	*k_x_*_2_/m^−1^(FEA)	Error-1	Error-2
0.01	6.65	6.60	6.37	4.36%	3.68%
0.02	13.27	12.93	12.45	6.62%	3.83%
0.03	19.85	18.69	18.30	8.50%	2.12%
0.04	26.36	23.69	23.94	10.11%	1.04%
0.05	32.78	27.70	29.36	11.66%	−5.67%

**Table 6 polymers-12-01001-t006:** Principal curvature of anti-symmetric cylindrical shells with different planform shapes from analytical model and simulation.

Planform Shape	*k_x_*_2_/m^−1^(EUCM)	*k_x_*_2_/m^−1^(NUCM)	*k_x_*_2_/m^−1^(FEA)	Error-1	Error-2
RPS	26.36	23.69	23.94	10.11%	−1.04%
EPS	26.30	23.98	24.46	7.52%	−1.96%
TPS	26.54	24.48	22.16	19.76%	10.47%

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
