# Peer review of "Non-Uniform Curvature Model and Numerical Simulation for Anti-Symmetric Cylindrical Bistable Polymer Composite Shells"

_polymers, 2020, doi:10.3390/polym12051001_

Round 1
Reviewer 1 Report
Please refer to the following comments:
- Page 7, line 188, xz and yz planes were not determined in Fig. 3. Please add a coordinate system in this figure.
- In line 189, in the sentence ended to “……as indicated in Fig. 3(b)” do you mean Fig. 3(a)? Because the four-point loading configuration is shown in Fig. 3(a).
- In the two-point loading method, what is the reason of using the rigid indenter instead of applying displacement at the respective nodes?
- Lines 195 and 196: “The displacement loadings are removed with the option Nlgeom on, and stabilize off to avoid inaccuracies.” If the displacement loading is removed, which load is applied? Please explain.
- In this study, the authors compared the results obtained from a new generated analytical model (QVCM (5DOFs)) with a non-validated FE simulation results. How did the authors validate the FE simulation? If the FE Simulation is not validated with experimental or analytical results, then it could not be applied to the validation of the proposed analytical model.
- The conclusion is very general. Please specify the accuracy of the method by quantifying the comparisons. For example, in the expression “The predictions of the presented model correlate fairly well with the simulation results….” specify the difference with a quantity or a percentage.
Author Response
We thank you very much for your positive comments. We have tried our best to revise the manuscript according to your comments and the changes have been marked in red, please see the attachment for a point by point response to your comments.
Reviewer 2 Report
Suggestions:
1) Analytical model: Pag 2 and 3. These two pages present a detailed mathematical equations that does not allow the reader. It is suggested to prepare an appendix for maths-equation and summarise in Analytical Model only the methodology adopted with the main points.
2) Please cut fig. 2, it is not necessary, and report in a table all the main parameters adopted. This part of the paper is similar to a technical report. Please, short this part it is too long and the real focus is not clear.
3) 4.4.2:The is "Influence of material properties", please add a table with the material type and properties. Otherwise change the title, because you are discussing different configurations.
4) introduce and better describe the meaning of: QVCM, IUCM, EUCM
Rewrit ethe Conclusion. There i sany Conclusion, try to foculise the main results obtained and more other main points to the introduction and discussion. i.e: this is not a Conclusion: "The effects of various factors including the angle of layup α, the 352 number of plies p, the angle of embrace γ, the longitudinal length l, the initial natural curvature h0 353 and the planform shapes are studied." it is an introduction!
Author Response

(The authors gave the same response as above.)

Round 2
Reviewer 1 Report
The author responded to all of the comments. The manuscript text should be refined regarding the English language.
Author Response
We thank you very much for your positive comments. We have tried our best to revise the manuscript according to your comments and the changes have been marked in yellow, please find the attachment for a point by point response to your comments.
Reviewer 2 Report
The paper has been improved, Thank you, but still remain several gray points, easily to be answered.
Line 269: I cannot found any clear error evaluation ot table reporting for: The discrepancy between results of QVCM and FEA are approximately within 1%.
Line 281: describe where I can easily check this: “The results of NUCM and FEA agree well with error below 5% and show that the principal curvature kx2 with the increase of γ.” (i.e.: as it is possible to realise from …fig… ot table…)
Line 302: Put the references (table… or figure…) for: A sensible increment of the principal curvature is observed for numerical results and results obtained by EUCM and NUCM.
Line 320: You state that: The accuracy of NUCM is still validated. Explain using references to the Tab. 4 data… or other results you need for.
Tab.4: present the results and discuss the differences obtained, explaining the reasons. The table has not discussed in details.
Line 345: Quote a table or a result you have found to state this: Theoretically the principal curvatures of anti-symmetric cylindrical shells with difference planform shapes are close, which show a certain error with FEA result for TPS. Why?.
Author Response

(The authors gave the same response as above.)
